# Beyond Implicit Bias: The Insignificance of SGD Noise in Online Learning

## Abstract

The success of SGD in deep learning has been ascribed by prior works to the *implicit bias* induced by high learning rate or small batch size ("SGD noise"). While prior works that focused on *offline learning* (i.e., multiple-epoch training), we study the impact of SGD noise on *online* (i.e., single epoch) learning. Through an extensive empirical analysis of image and language data, we demonstrate that large learning rate and small batch size do *not* confer any implicit bias advantages in online learning. In contrast to offline learning, the benefits of SGD noise in online learning are strictly computational, facilitating larger or more cost-effective gradient steps. This suggests that SGD in the online regime can be construed as taking noisy steps along the "golden path" of the noiseless *gradient flow* algorithm. We study this hypothesis and provide supporting evidence in function space by conducting experiments that reduce SGD noise during training and by measuring the pointwise functional distance between models trained with varying SGD noise levels, but at equivalent loss values. Our findings challenge the prevailing understanding of SGD and offer novel insights into its role in online learning.

## 1 Introduction

In the field of optimization theory, the selection of hyperparameters, such as learning rate and batch size, plays a significant role in determining the *optimization efficiency*, which refers to the computational resources required to minimize the loss function to a predetermined level. In strongly *convex problems*, altering these hyperparameters does not affect the final solution since all local minima are global. Hence, the final model only depends on the *explicit biases* of architecture and objective function (including any explicit regularizers). In contrast, Deep Learning is *non-convex*, which means that the choices of algorithm and hyperparameters can impact not only optimization efficiency but also introduce an *implicit bias*, i.e., change the regions of the search space explored by the optimization algorithm, consequently impacting the final learned model.

The implicit bias induced by the algorithm and hyperparameter choices can significantly affect the quality of the learned model, including generalization, robustness to distribution shifts, downstream performance, and more. Hence, the implicit bias of *stochastic gradient descent (SGD)* has garnered considerable attention within the research community (Jastrzebski et al., 2017; Lewkowycz et al., 2020; Damian et al., 2021; He et al., 2019; Nacson et al., 2022; HaoChen et al., 2021; Andriushchenko et al., 2022). In part, this implicit bias emerges due to SGD using a *noisy approximation* of the population gradient. This noise arises from using a minibatch to estimate the true gradient, and using a non-negligible learning rate, leading to a deviation from the linear approximation of the loss.[1]

Perhaps counter-intuitively, SGD noise is often deemed *advantageous* for implicit bias. In particular, several works showed that higher learning rates and smaller batch sizes yield *flatter* minima (Keskar et al., 2016; LeCun et al., 1998; Masters & Luschi, 2018; Goyal et al., 2017), which tend to generalize well (Hochreiter & Schmidhuber, 1997; Smith et al., 2020) (see also Figure 1, top). However, these works are limited to the setting of multi-epoch or *offline training*.

---

[1]In this work, *SGD noise* corresponds to the noise introduced by using a non-infinitesimal learning rate and a finite minibatch size. Unlike some works, we do *not* use "SGD Noise" to refer to the randomness induced by the random shuffling of datapoints.

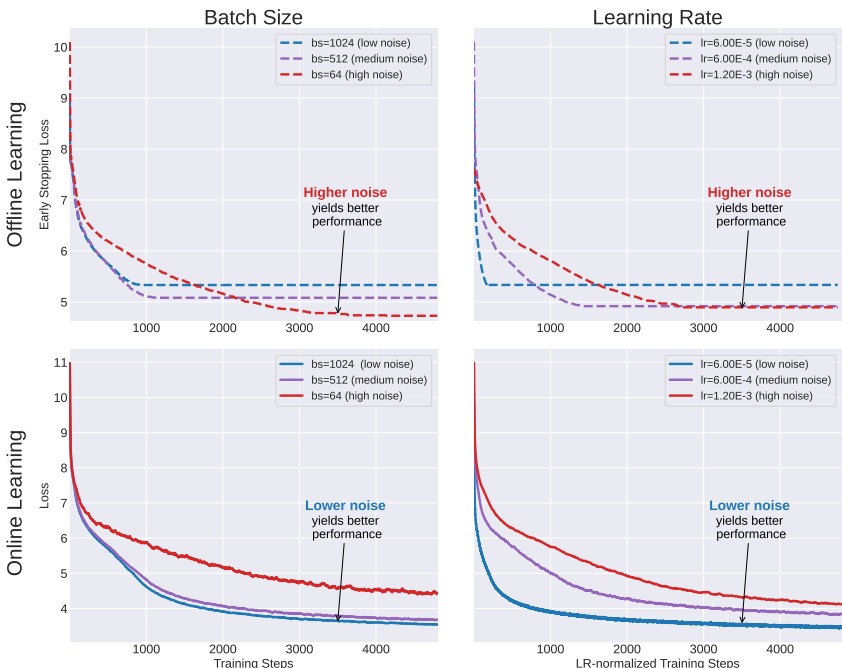

Figure 1: Experiments on offline (**top row**) and online (**bottom row**) learning on the C4 dataset across various batch sizes (**left column**) and learning rates (**right column**). As shown in prior works, in *offline learning* (top row), higher SGD noise (lower batch size or larger learning rate) offers an implicit bias advantage and plateaus at a lower loss. In contrast, we show that in *online learning* (bottom row), higher SGD noise does not provide any implicit bias benefit to performance, and lower noise reaches a smaller loss. On the top row, the y-axis measures early stopping (true) loss. In the right column, training steps on the x-axis are scaled by the learning rate in order to enable more accurate comparisons between different learning rates. See Section 2 for more details.

In this work, we examine the implicit bias of SGD in the **online learning** setting, in which data is processed through a *single epoch*. Online learning is common in several self-supervised settings, including large language models (LLMs) (Komatsuzaki, 2019; Brown et al., 2020; Hoffmann et al., 2022b;a; Chowdhery et al., 2022). While in online learning, the train and test distributions are identical (and hence there are no generalization considerations), it is still a *non-convex* optimization. So, the inductive bias introduced by algorithm and hyperparameter choices could still potentially play a major role in learning trajectory and model quality. However, we find that the impact of SGD noise parameters in practical settings of online learning is qualitatively similar to their impact on *convex* optimization. Specifically, we undertake an extensive empirical investigation and find that, *in online learning, SGD noise is indeed only "noise"* and offers no implicit benefits beyond optimization efficiency. This can be seen in Figure 1 (bottom row), where we observe that (neglecting computational cost) performance in online learning improves with increasing batch size and decreasing learning rate.

**The "Golden Path" Hypothesis**. When taking SGD to the limit of large batch size and small learning rate we get the *noiseless Gradient Flow* (GF) algorithm (Bach, 2020), wherein each step consists of an infinitesimal movement in the direction of the population gradient. By the above discussion, ignoring computational constraints, Gradient Flow is the optimal method in the context of online learning. Our findings hint that the SGD path is just a noisy version of the underlying noiseless GF path as illustrated in Figure 2. We propose this conjecture as the *golden path* hypothesis, which stands in contrast to the alternative "Fork in the Road" possibility, wherein SGD and GF discover qualitatively different functions. As shown in prior works (as well as in our own experiments), the "fork in the road" scenario (and specifically scenario a1, where high noise leads to a better minima) is the typical case for *offline learning*.

To be precise, our work gives evidence for the following hypothesis:

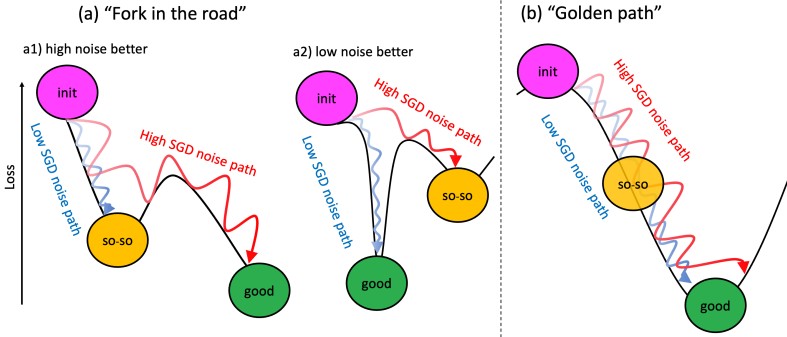

Figure 2: A-priori online learning can exhibit two potential scenarios: **(a)** *"Fork in the Road,"* wherein the selection of learning rate and batch size leads the optimization algorithm to explore distinct regions of the search space, potentially resulting in different loss outcomes. (a1: better loss for the high-noise path, which is the common case for offline learning, and a2: better loss for the low-noise path). **(b)** *"Golden path,"* wherein the optimization trajectory remains similar for both gradient flow and SGD. In the latter scenario, the noise in SGD primarily influences the algorithm's traversal speed (and stability) along the path. Our research provides evidence supporting the "golden path" scenario for online learning.

---

> **Golden Path Hypothesis**: For natural settings in online learning, a path trained via SGD does not deviate far from one trained via GF, in the following sense:
>
> - **Loss Trajectories**: If SGD noise is dropped[2] from high to low at time $t$, then shortly after $t$, the loss curve will "snap" to track the curve of a model that was trained with low noise from initialization, and hence followed the "golden path".
>
> - **Function Space**: After reducing the SGD noise, the resulting function from this path is also similar in functional distance to that of a path with low noise from initialization.

**Is the Hypothesis False?** Indeed, the "Golden Path" hypothesis does not hold in a variety of settings. One simple toy example is provided in Figure 2 of Wu et al. (2018), demonstrating how higher noise could help in escaping "sharp" minima in the non-convex regime (for further discussion, refer to the related works). However, we claim that in "natural settings"—real world datasets trained using deep learning—we observe that the SGD path becomes "close" to that of the GF path once the SGD noise has been reduced. As we have stated above, closeness is measured both in terms of the loss trajectories and functional distance. One could also consider a stronger notion of the golden path hypothesis in weight space; however, in its naive formulation, the golden path hypothesis **does not hold in weight space**; this is due to the presence of permutation symmetries, dead neurons, and structural differences (see section 1.1). While we believe that a more careful formulation of the "golden path" hypothesis can hold in weight space, we focus on loss and function space in this paper and defer weight space exploration for future work.

Overall, our work gives evidence to the hypothesis that there is a "noiseless" or "golden" path that Gradient Flow takes, and that the learning rate and batch size hyperparameters play no role in the choice of the path but only in the computational cost to travel on it, the training stability as well as the level of "variance" along the path. Hence choosing these parameters should be determined by balancing their negative impact on noise with their positive impact on computation. This is in stark contrast to the role of SGD in *offline learning*, wherein SGD noise can influence not just the *speed* of optimization but also its *journey* and even its final *destination* (i.e., function at convergence).

**Contributions and organization.** We delineate our contributions as follows:

1. Our first contribution is demonstrating that, unlike in offline learning, SGD noise does not provide any implicit bias advantage in a variety of practical online learning settings. This is presented in Section 2, which contains a systematic investigation of the effects of SGD noise

---

[2]Note that high SGD noise runs have more variance, and to remove this factor we equalize the noise of two trajectories before comparison. See Figure 7 for an example.

    in the online versus offline settings. Our analysis encompasses both vision (ResNet-18 on CIFAR-5m, ConvNext-T on ImageNet) and language tasks (GPT-2-small on C4),

2. A second contribution is to propose and examine the "golden path" hypothesis in the context of online learning. In Section 3, we provide evidence that SGD loss trajectories follow that of gradient flow by showing that the loss curves of high-noise SGD "snap" to those of low-noise SGD when the noise levels are equalized.

3. In Section 4, we further substantiate the "golden path" hypothesis in *function space*. We present evidence that models trained with varying levels of SGD noise learn the same functions, indicating that the differences in noise do not significantly impact the learned representations.

Overall, our work sheds new light on the roles of batch size and learning rate in online deep learning, showcasing that their benefits are merely computational. We also provide a pathway for a more unified understanding of training trajectories, by giving evidence that SGD takes noisy steps that approximate the "golden path" that is taken by gradient flow.

## 1.1 RELATED WORK

**Implicit Bias:** A considerable volume of literature has been devoted to examining the impact of learning rate and batch size on the training of neural networks from both theoretical (Andriushchenko et al., 2022; Lewkowycz et al., 2020; He et al., 2019; Nacson et al., 2022; Paquette et al., 2022a;b) and practical (Geiping et al., 2021; Nado et al., 2021; Masters & Luschi, 2018; Xing et al., 2018; Jastrzebski et al., 2020a; Karpathy, 2019) perspectives. Among practitioners, the consensus revolves around maximizing computational resources: large batch sizes are employed to fully exploit the hardware, while the learning rate is scaled with the batch size to maintain optimization stability (You et al., 2019). However, regarding optimal hyperparameters, it is widely held that higher learning rates and smaller batch sizes result in superior minima (Keskar et al., 2016; LeCun et al., 1998; Masters & Luschi, 2018; Goyal et al., 2017). Although some empirical studies (Geiping et al., 2021; Lee & Avestimehr, 2022; Novack et al., 2022; Hoffer et al., 2018) contest this notion by utilizing various techniques, it remains the prevalent intuition within the community. From a theoretical standpoint, several works showed the benefit of SGD noise (i.e., a higher learning rate and smaller batch size) as yielding a more favorable implicit bias (Damian et al., 2021; HaoChen et al., 2021; Ali et al., 2020; Blanc et al., 2020; Jastrzebski et al., 2020b). These works show that in certain overparameterized settings, higher SGD noise leads to a better generalization.

Within non-convex optimization literature, it is known that SGD can exhibit important optimization effects such as escaping saddle points and sharp minima (Kleinberg et al., 2018). A simple example of such escape behavior is provided in Figure 2 of Wu et al. (2018). Using similar analysis, Li et al. (2019) showed that, under certain "simplicity bias" settings, initial high learning rate can help in avoiding learning "easy-to-generalize" features.

In convex optimization, however, diminishing the stochastic gradient descent (SGD) noise typically leads to enhanced performance, even when accounting for LR-normalization. For instance, Paquette et al. (2022a;b) establish that, under certain assumptions for high-dimensional random features models, both higher learning rate and a reduced batch size result in a worse test error for a given value of LR-normalized training steps. These works operate within a regime where number of data points scales proportionally with the model size, thereby aligning more closely with the "online learning" paradigm. Nonetheless, this relationship does not universally hold, as demonstrated by a counterexample presented in Appendix E, which depicts an edge case situation where a smaller learning rate results in a "slower" convergence rate.

**Offline vs. Online:** One distinction absent from the aforementioned discussion is the difference between the online and offline regimes. For instance, Smith et al. (2021) clearly investigate the effect of a large learning rate for a single epoch training, and show that SGD with high learning rate has an implicit bias towards reducing gradient norms. In contrast, we observe empirically that in the online setting, implicit bias doesn't affect the network in function space. The Deep Bootstrap framework of Nakkiran et al. (2021); Ghosh et al. (2021) contrasts the online and offline worlds, revealing that a significant portion of offline training gains can be attributed to its online component. Recent works also demonstrated the detrimental effects of repeating even a small fraction of data (Hernandez et al., 2022; Xue et al., 2023), for LLMs. See Appendix B for additional related works.

## 2 THE IMPLICIT BIAS OF SGD IN ONLINE LEARNING

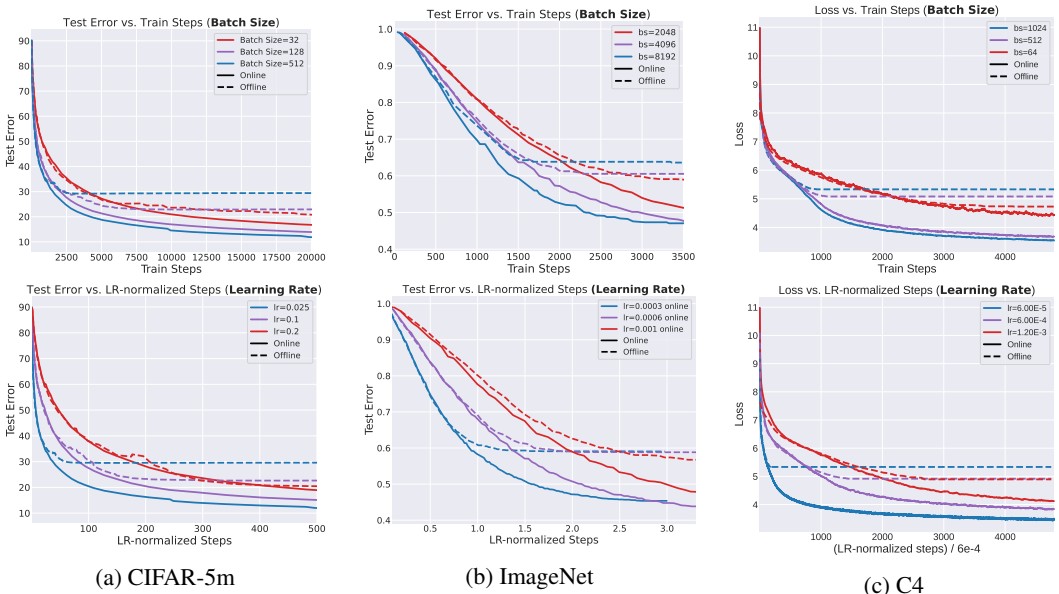

(a) CIFAR-5m       (b) ImageNet       (c) C4

Figure 3: Test performance for ResNet-18 trained on CIFAR-5m (**left**), ConvNext-T on ImageNet (**middle**), and GPT-2-small on C4 (**right**) across varying batch sizes (**top**) and learning rates (**bottom**). For the bottom row, training steps on the x-axis have been scaled by the learning rate. Red corresponds to high SGD noise (high LR or small batch), blue to low SGD noise (small LR or high batch), and purple to an intermediate setting. Solid (resp. dotted) lines correspond to runs in the online (resp. offline) setting. For online learning, lower SGD noise runs consistently outperform higher noise runs per given step (or normalized step in case of LR). Offline learning performance initially matches online performance, eventually runs with higher noise outperform low-noise runs. All experiments are averaged over $\geq 4$ runs. See Figure 8, 9 for error bars and more hyperparameter values.

In this section, we present our experimental results on the impact of SGD noise (i.e., magnitude of learning rate and size of batch) on implicit bias. We show that the effect of this noise can differ significantly between the *offline* and *online* regimes. Since our goal is to study the impact of SGD noise on implicit bias rather than on computational efficiency, in our batch-size experiments, we measure loss as a function of the number of gradient steps, and not as function of FLOPs. Similarly, when varying learning rate, we need to account for the fact that a larger step size also corresponds to "more movement" in parameter space. Thus, in our learning-rate experiments, we *rescale* the number of steps by the learning rate, employing the formula: `LR-normalized steps = steps x LR`. When the learning rate varies, we scale each step separately, using $\sum_{i=1}^{t} \eta_i$ as this measure, where $\eta_i$ is the learning rate used in step $i$.

We conduct an experimental evaluation of our claims employing convolutional models in computer vision and Transformer models in natural language processing (NLP). Specifically, we run ResNet-18 on CIFAR-5m (Nakkiran et al., 2021), a synthetically generated version of CIFAR-10 with 5 million examples, ConvNext-T on ImageNet, and GPT-2-small on C4. To imitate the online regime with ImageNet, we only train for 10 epochs with data augmentation. As we show in Figure 3, we find that

1. In the **offline setting**, consistent with prior work (Keskar et al., 2016), SGD noise can (and often does) lead to better implicit bias for the final models. Specifically, even if runs with smaller noise initially[3] decrease the loss faster, eventually they get "stuck" at a worse local minima than the runs with higher SGD noise (larger learning rate or smaller batch size). This is consistent with Scenario a1 of Figure 2 ("fork in the road" with high noise being better), where a higher noise enables escaping from bad local minima.

---

[3]The curves for offline learning initially track the online learning curves (as predicted by Deep Bootstrap (Nakkiran et al., 2021)) but then plateau at a higher loss for lower SGD noise.

2. In contrast, in the **online setting**, the implicit bias advantage of SGD noise *completely disappears*, and the main benefit from large learning rate and small batch sizes reduces to being just computational. Specifically, after we control for computation (either by measuring gradient steps for varying batch experiments, or measuring normalized steps for varying learning-rate experiments), the low-noise runs consistently outperform the higher noise runs. This is consistent with either Scenario a2 of Figure 2 ("fork in the road" where a lower noise run can explore better minima) or with Scenario b (the "golden path": higher noise follow a similar trajectory but with some degradation due to noise). As we show in Sections 3 and 4, our additional experimental results give evidence for the latter (i.e., "golden path") case.

**Isn't this trivial?** One natural intuition is that regularization is not needed in the online learning setup because there is no difference between optimization and generalization (or between train and test). However, we have empirically observed that well-known explicit regularization techniques such as weight decay can yield better performance in the online regime for standard tasks (Figure 10, Appendix F.1), suggesting that this inuition might be flawed and more nuance is required.

We show further experiments in Appendix F.1 including an experiment with low SGD noise where the loss trajectory has nearly converged to that of gradient flow. See Appendix A for more details.

## 3 SNAPPING BACK TO THE GOLDEN PATH

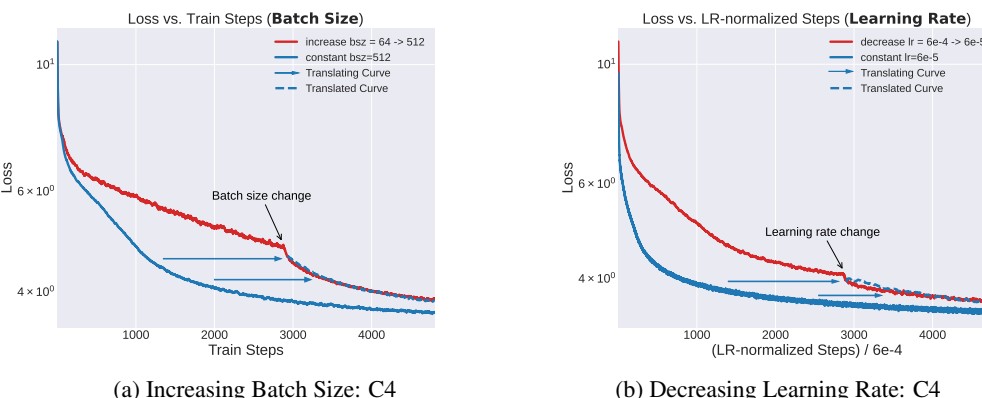

(a) Increasing Batch Size: C4  (b) Decreasing Learning Rate: C4

Figure 4: Decreasing SGD noise (**left**: batch size, **right**: learning rate) during training for the C4 dataset and GPT2-small. The red curves correspond to models trained with low SGD noise from initialization, and the blue curves correspond to models trained with high SGD noise and then dropped after $T_0$ steps. Across all experiments, dropping SGD noise causes the blue curves to follow a translated version (dashed red) of the low-SGD noise model curve.

The results of Section 2 show that SGD noise does not benefit implicit bias in online learning. As we discussed in Figure 2, there are two potential explanations for why in online learning (unlike the offline case), decreasing SGD noise steers optimization towards a smaller-loss trajectory. One explanation is Scenario (a2) of the figure. Namely, it may be the case that choosing low SGD noise leads the optimization algorithm to a different (and better) trajectory, that is completely inaccessible to the high SGD-noise runs. The second is the "golden path" hypothesis: higher-noise runs travel on approximately the same path as lower-noise ones, suffering some loss-degradation resulting from the imperfect approximation. To rule out the first explanation, we conduct the following experiment in the online setting (see Figure 4, left):

1. Run two experiments—one with high batch size, and one with small batch size—for $T_0$ steps.

2. After $T_0$ steps, decrease the SGD noise by increasing the batch size of the second experiment to match the hyperparameters of the first one, and continue both runs.

Under the golden path hypothesis, we expect that shortly after increasing the batch size (i.e., at $T_0 + \tau$ for $\tau \ll T_0$), the loss curve would "snap" to the golden path, and continue following the same

trajectory of the model that was trained with low SGD noise. On the other hand, the (a2) scenario of Figure 2 implies that decreasing the noise would not result in any significant change to the loss curve.

We perform a series of experiments as described above with GPT2-small on the C4 dataset in Figure 4 (See Figure 12 for CIFAR-5m). In addition to the batch size experiments we conduct analogous experiments with a sudden decrease in LR. We consistently observe that, after dropping the SGD noise at some time $T_0$, the loss sharply improves to some value $\ell_0$. From this point onward, the loss curve of the model is nearly identical to a right-translation of the loss curve of the model that was trained with low SGD noise from initialization. This phenomenon does not hold generally in offline learning, as it is well known that at convergence different minima are reached by gradient flow and SGD.

## 3.1 Increasing SGD Noise

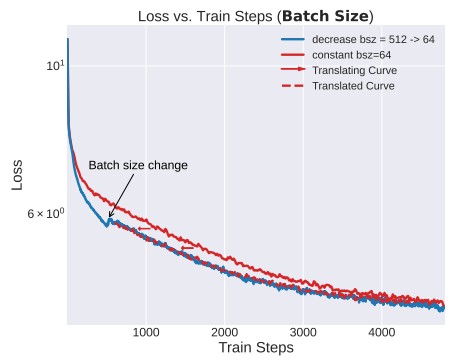
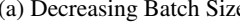
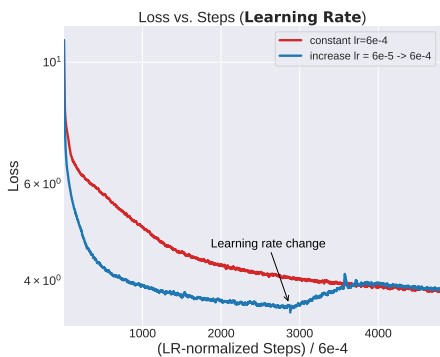

(a) Decreasing Batch Size       (b) Increasing Learning Rate

Figure 5: Loss dynamics for GPT-2-small on the C4 dataset upon increasing the SGD noise at time $T_0$. The left panel shows the effect of decreasing the batch size , while the right panel demonstrates the impact of increasing the learning rate. The loss snapping phenomenon is strongly observed in the learning rate experiment, essentially requiring no translation. In the case of batch size, we see a similar phenomenon as in Figure 4 where following the increase in SGD noise, the loss curve of the blue line follows a translated version of the loss curve of the run with a lower batch size.

In contrast to exploring the transition from a noisy trajectory towards the "golden path", we can also investigate the effects of introducing SGD noise, thereby deviating away from the optimal trajectory. To this end, we conduct an analogous experiment to the one presented in Section 3, but with a crucial difference: instead of reducing the noise in SGD at time $T_0$, we *increase* it, for instance, by increasing the learning rate or decreasing the batch size. Looking at the results for GPT-2-small on the C4 dataset, we observe an immediate and significant increase in the loss upon introducing additional noise, as illustrated in Figure 5 for both the batch size decrement (left) and learning rate increment (right) scenarios. Interestingly, the loss snapping phenomenon is strongly evident in the case of the learning rate experiment, where following the increase in the learning rate, the loss curve rises to exactly match the loss curve trained with this high learning rate from initialization. In the batch size reduction experiment we see a similar phenomenon as in the experiments shown in Figure 4, where the lower noise loss curve, after noising, follows a translated version of the higher noise curve. We conjecture that the differences in shifts is due to the additional noise causing some progress on the path to be lost (and then needs to be recovered). See Appendix D for further discussion.

## 4 Pointwise Prediction Differences Between Trajectories

Section 3 offered empirical substantiation for the golden path hypothesis, but was restricted to claims about the *loss curve*. Now we investigate a stronger version of this hypothesis: the golden path in **Function Space.** Specifically, our goal is to verify whether trajectories using different SGD noise are functionally similar. Due to differing noise levels between the trajectories, we cannot directly compare their functional similarity; instead, we show that lowering the SGD noise causes models to "snap" to the golden path in a functional sense, i.e. when measuring the *pointwise* distance.

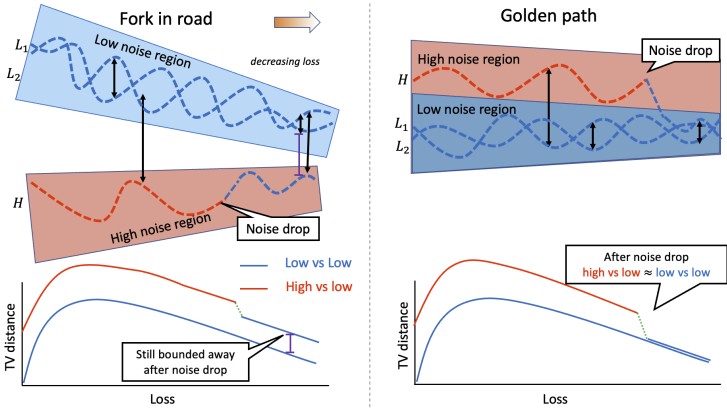

Figure 6: A schematic representation of the potential total variation distance in function space under the "fork in the road" (left) and "golden path" (right) scenarios. In the "fork in the road" scenario, low and high learning rates traverse distinct regions of the function space, resulting in a persistently high total variation distance even when the learning rate is decreased. Conversely, in the "golden path" scenario, the high learning rate follows an approximate or "noisy" version of the low learning rate trajectory in function space. Consequently, reducing the learning rate causes the trajectory to "snap" to the low learning rate path, achieving a reduced total variation distance compared to two independent low learning rate trajectories.

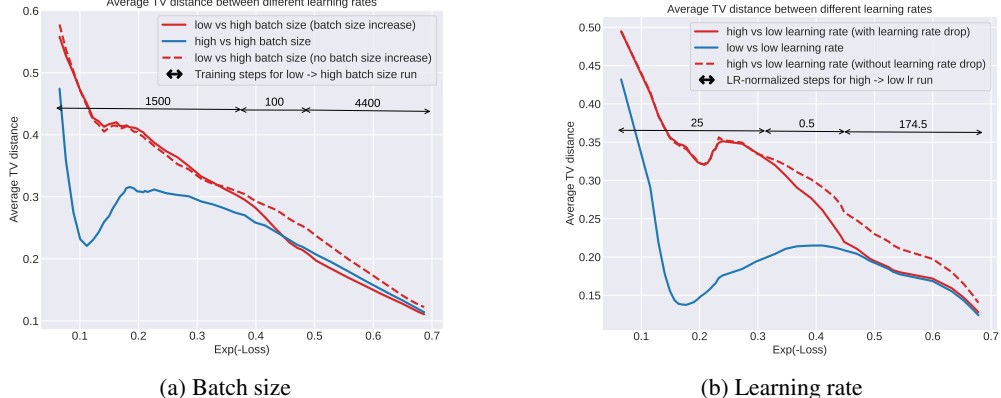

(a) Batch size
(b) Learning rate

Figure 7: Average TV distance behavior on CIFAR-5m with and without reducing SGD noise. On the left panel, for one run, we increase the batch size from 32 to 512 and on the right panel we reduce the learning rate from 0.05 to 0.005. The dashed line in both the curves represents the run with constant high SGD noise as a baseline. The figure demonstrates that the distance between the model with reduced noise and a low-noise model at the same loss is nearly identical to the distance between two random low-noise models trained with the same hyperparameters. This is in contrast with the distance between a high noise and a low-noise model at the same loss. This observation supports the "golden path hypothesis" and contradicts the "fork in the road" hypothesis.

To be precise, we investigate the functional distance between models by measuring the average total variation (TV) distance[4] of their softmax probabilities on the test dataset. Figure 6 gives a schematic illustration of our experimental setting. In this model, we have a low noise SGD run to serve as the "ground truth" for the golden path. The blue curve in Figure 6 serves as a baseline for the functional distance (TV) to this "golden path" as it represents another low noise run trained from a different initialization. This is a strong baseline since different initialization seeds are often treated as a nuisance parameter in deep learning. We compare it to the TV distance of the high noise path depicted by the red curve.

---

[4]The TV distance is defined as the half of $\ell_1$ distance for probability measures.

As illustrated in Figure 6, under the "fork in the road" hypothesis, we expect that high noise and low noise trajectories will explore different regions of the search space. Thus, even if dropping the noise improves loss, the baseline (distance between independent runs of low SGD noise) will still be significantly lower than the distance between the 'dropped-noise model' and the low-noise model.

In this section, by an empirical study on CIFAR-5m, we show that this is not the case, and TV distance behaves as would be expected from the "golden path hypothesis". Specifically, as shown in Figure 7, after dropping the noise, the TV distance between this model and a low noise model at the same loss is virtually identical to the TV distance between *two random low-noise models trained with identical hyper parameters*. Figure 7 also shows that this is not true for the high noise model, demonstrating that the two paths indeed differ due to the difference in the level of noise. Moreover, in Figure 7, it could also be seen that the high noise model reaches the "golden path" very quickly (in terms of LR-normalized steps) upon dropping the SGD noise.

As shown in Figure 13, after dropping the learning rate, the TV distance between a model trained with the learning rate decay and the low noise model becomes comparable to that of two independent low-noise models trained with identical hyperparameters. See Appendix F.3 for additional experiments including those on the C4 dataset.

## 5 DISCUSSION AND CONCLUSION

The results of our investigation reveal a striking discrepancy between the online and offline learning regimes, in terms of the implicit bias of SGD. In the offline regime, this bias exhibits a beneficial regularization effect, whereas in the online regime, it merely introduces noise to the optimization path. This critical distinction between the two regimes has largely been overlooked in both theoretical and empirical research, with few studies explicitly addressing the difference. We argue that recognizing and accounting for the online versus offline learning regimes is crucial for understanding various deep learning phenomena and for informing the design of optimization algorithms.

Although our work represents only an initial exploration into the disparities between the online and offline learning regimes, we can draw several immediate conclusions.

**Implications for Practitioners.** For online learning, our findings emphasize the relative *simplicity* of hyperparameter tuning, primarily focusing on computational efficiency and stability. In situations where data or computational resources are limited, however, the regularization effects of SGD become more significant, and hyperparameter selection and optimization take on greater importance. For instance, in the low-data regime, it may be crucial to use a smaller batch size, even if results in not fully utilizing the GPU. In online training, this consideration appears to be consistently irrelevant.

**Implications for Theorists.** The primary takeaway for theoretical research is that the study of the regularizing effects of high SGD noise should be confined to the offline learning regime, as failing to make this distinction creates tension with practical applications. Theoretical findings that do not account for this difference can not fully capture why SGD is effective for deep learning. Furthermore, our observation that SGD follows a noisy trajectory near the "golden path" of gradient flow in loss and function spaces, coupled with the Deep Bootstrap (Nakkiran et al., 2021) assertion that a substantial portion of offline training can be explained by the online regime, implies that gradient flow may be instrumental in understanding many aspects of deep learning.

In conclusion, given that many large-scale deep learning systems, such as Language Models (LLMs), predominantly operate within the online learning regime, our findings challenge the conventional understanding of deep learning, which is primarily based on offline learning. We contend that it is necessary to reevaluate our comprehension of various deep learning phenomena in the context of online settings. Moreover, we propose gradient flow as a promising theoretical tool for studying online learning, considering the minimal impact of SGD noise on the functional trajectory.

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

## A  EXPERIMENTAL DETAILS

### A.1  CIFAR-5M

In our CIFAR-5m experiments, we trained ResNet-18, on normalized (across channels) images and using the SGD optimizer with 0.9 momentum.

**Section 2:**  For Figure 3 (a) we trained without any data augmentations. For Figure 3 (a, top) we used a learning rate of 0.025 and for Figure 3 (a, bottom) we used a batch size of 512. For offline learning we trained on a random subset of 50k samples (class balanced). Both plots use exponential moving average with coefficient of .8.

**Section 3:**  For Figure 4 (a) we trained with standard CIFAR data augmentation of a random crop (`RandomCrop(32, padding=4)` in Pytorch) and horizontal flip (`RandomHorizontalFlip()` in PyTorch). For Figure 12a we used a learning rate of 0.025 and for Figure 12b we used a batch size of 512.

**Section 4:**  For Figure 7, we trained on CIFAR-5m without any augmentations so as to remove the pointwise difference due to different augmentations. For (a), we trained two networks with learning rate 0.05, one with batch size 32 and the other with 512, both for 12000 steps ($\sim 1$ epoch for batch size 512) and the one with batch size 32 was changed to 512, after 1500 training steps. For (b), we trained two networks with batch size 128, one with a constant learning rate of 0.005 and one with a step decay from 0.05 to 0.005 at 500 steps. Both of these were trained for LR-normalized steps ($\sum \eta_i$, where $\eta_i$ is the learning rate at step $i$) of 40000 ($\sim 1$ epoch of 0.005).

### A.2  IMAGENET

For the ImageNet experiments, we used ConvNext-T Liu et al. (2022) and unless specified otherwise, used a batch size of 2048 and learning rate of $1e$-$4$ with the AdamW optimizer with weight decay 0.005. For all experiments, we use cosine decay scaling of the learning rate with respect to training steps (not epochs). We used the `RandomHorizontalFlip` and `RandomCrop` augmentations and also preprocessed the dataset to be resized to $256 \times 256$ using OpenCV before training for speed purposes. For the offline results, we downsample the dataset by a factor of 10, i.e., use 128k examples.

### A.3  C4

For all experiments we trained GPT-2-small (124m parameters) on the C4 dataset using the codebase from Mosaic ML ML (2023). Hyperparameters—aside from learning rate, batch size, and training duration— equal the default values by Mosaic ML [5]. For learning rate and training duration, the default values are $6 \times 10^{-4}$ and 4800 training steps respectively, and hence for all experiments we maintain this ratio (e.g., for learning rate $6 \times 10^{-5}$, we train for 48000 steps). All plots are generated using an exponential weighted moving average after logging at every step.

**Section 2:**  For Figure 3 (**c, top**) we used a learning rate of $6 \times 10^{-4}$ and for Figure 3 (**c, bottom**) we used a global batch size of 256. For offline learning we trained on a random subset of roughly 100 million tokens, and all hyperparameters are otherwise identical with the analogous run in the online setting.

**Section 3:**  In Figures 4 and 5, we change the learning rate from $6 \times 10^{-4}$ to $6 \times 10^{-5}$ and vice versa, and respectively change the batch size from $64$ to $512$ and vice versa. When changing the learning rate we keep the global batch size fixed to 256, and when changing the batch size we keep the learning rate fixed to $6 \times 10^{-4}$. For Figure 5 (**left**), the change occurs after 500 training steps; for the other plots in Figures 4 and 5, the change occurs after 60% of the training duration has elapsed (i.e. after 2880 (LR-normalized) steps). To prevent learning instabilities when suddenly increasing the learning rate in Figure 5 (**right**), we increase the learning rate linearly over 100 (LR-normalized) steps.

---

[5]`https://github.com/mosaicml/examples/blob/main/examples/llm/yamls/`
`mosaic_gpt/125m.yaml`

## B ADDITIONAL RELATED WORK

**Network Evolution.** Similar to us, multiple works discuss the similarity of SGD dynamics across hyperparameter choices. This question has been studied from the lenses of example order (Nakkiran et al., 2019b; Shah et al., 2020; Hacohen et al., 2020; Baldock et al., 2021), representation similarity (Kornblith et al., 2019; Bansal et al., 2021), model functionality (Olsson et al., 2022), loss behaviour (Nakkiran et al., 2019a), weight space connectivity (Fort et al., 2020; Frankle et al., 2020) and the structure of the Hessian (Cohen et al., 2021). Our work focuses on the online regime, and as opposed to previous studies, gives evidence to the *Golden Path* conjecture in this regime wherein SGD noise strides (in function space) along the gradient flow trajectory but with noise. As shown by both our work and others, the golden path conjecture does *not* hold for offline learning.

## C DISCUSSION OF RESULTS IN SECTION 2

In Figure 3, we showed the stark contrast in offline vs online learning regarding their interaction with SGD noise. For the offline setting, higher SGD noise generally leads to a better performance, while in online learning, SGD noise only hurts performance.

However, upon closer observation, we can see that in all the plots of Figure 3, offline learning performance closely follows the online learning performance throughout the majority of the training period before reaching a plateau. This is in agreement with the results of Nakkiran et al. (2021). They empirically demonstrated that, across a variety of scenarios, a major part of offline training can be explained away by online learning.

This result, combined with our observation about the SGD trajectory being a noisy version of the "golden path" exhibited by gradient flow in online learning, shows that gradient flow is a useful theoretical tool even for studying offline learning. In particular, given that in practice, we choose hyperparameters to maximize test performance, this means that we move as far along the online trajectory as possible. The "denoised" version of this online trajectory is given by gradient flow on the population loss. Thus, our work proposes studying "gradient flow on population loss" as an alternative (or a "denoised" version) of studying SGD in offline learning.

## D DISCUSSION OF RESULTS IN SECTION 3

As exhibited in Figure 5, we observe the loss snapping phenomenon even when we increase SGD noise during training. Intuitively, the model that initially has a low amount of SGD noise (high batch size/low learning rate) has progressed "far along" the path, and thus the additional noise results in progress lost on the path and an increase in loss. One plausible reason for this could be the recent phenomenon of Edge-of-Stability (Cohen et al., 2021), where the authors showed that SGD dynamically leads the model to an "edge-of-stability" curvature, i.e, increasing learning rate leads to an increase in loss in the successive steps. However, our paper shows that, following this brief increase in loss, the model is able to recover and continue training, albeit with higher SGD noise; thus the performance follows that of a noisier trajectory.

## E GRADIENT DESCENT ON LEAST SQUARES

In this section, we provide analysis of the simple case of gradient descent on least squares objective, where we show that, lower learning rate performs worse than higher learning rate, for a given value of LR-normalized training steps.

For simplicity, consider the 1-dimensional loss function given by $f(w) = \frac{1}{2}w^2$. For a given learning rate $\eta$, we know, at time $t$,

$$w_t = (1 - \eta)^t w_0$$

Clearly, using this, we can say

$$f(w_t) = (1 - \eta)^{2t} f(w_0)$$

If instead, we would have used a learning rate of $\frac{\eta}{2}$, then at time $2t$, the loss would be given by

$$f(\widetilde{w}_{2t}) = \left(1 - \frac{\eta}{2}\right)^{4t} f(\widetilde{w}_0)$$

Clearly, if both of these runs would have started from the same initialization point, then $f(\widetilde{w}_{2t}) \geq f(w_t)$ (this is true as $(1 - \frac{\eta}{2})^2 \geq (1 - \eta)$).

This provides a very simple case of gradient descent on quadratic loss, where for a given LR-normalized training steps, higher lr performs better than lower lr. This analysis is specifically for GD, for SGD we have to account for the stochasticity. We discuss this more in Section 1.1.

## F  ADDITIONAL PLOTS

### F.1  ADDITIONAL PLOTS FOR SECTION 2

In Figures 8 and 9 we show a subset of figures from Section 2 with error bars. Note that all figures in Section 2 were already averaged over $\geq 4$ runs.

In Figure 10 we jointly decrease learning rate and increase batch size and show that the loss curves converge (with the x-axis as LR-normalized steps) while maintaining the observation that lower SGD noise runs perform better.

In Figure 11 we show an example in which explicit regularization helps in online learning. This is unlike the implicit regularization of SGD noise which does not help in our experiments in online learning.

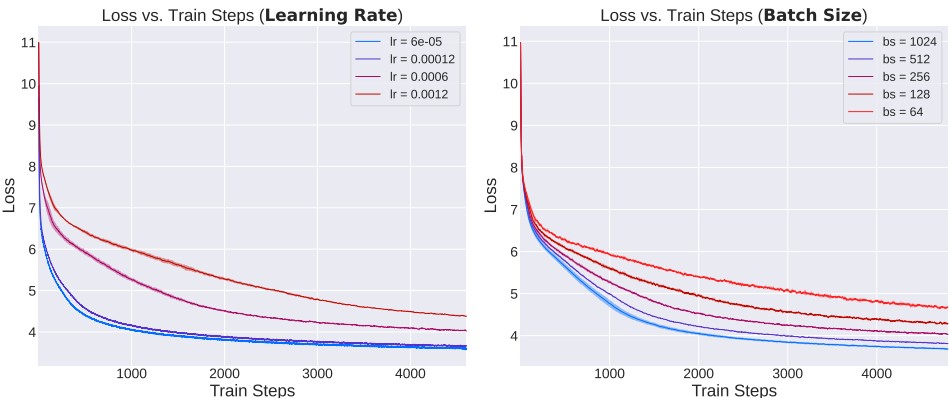

Figure 8: Experiments for online learning on the C4 dataset across several learning rates (**left**) and batch sizes (**right**) with error bars. Consistent with our findings, runs with a lower amount of SGD noise yield better performance.

### F.2  ADDITIONAL PLOTS FOR SECTION 3

In addition to the experiments for C4 in Section 3 we conducted a similar experiment for the CIFAR-5m dataset (Figure 12).

### F.3  ADDITIONAL PLOTS FOR SECTION 4

To further support the claim in Section 4, we perform the same experiment on the C4 dataset (Figure 13): after matching losses, we measure the average TV distance of the softmax probabilities on the validation set between models. One run which is trained with a constant learning rate of $6 \times 10^{-5}$ (low noise) serves as the "ground truth" for the golden path, and we measure the average TV distance to one run which is trained first using learning rate $6 \times 10^{-4}$ and decays to $6 \times 10^{-5}$ after 2880 LR-normalized steps (high $\to$ low noise). We compare this with the baseline, which measures the TV distance with another low noise run trained from a different initialization.

We also present additional results with different hyperparameters for CIFAR-5m in Figure 14.

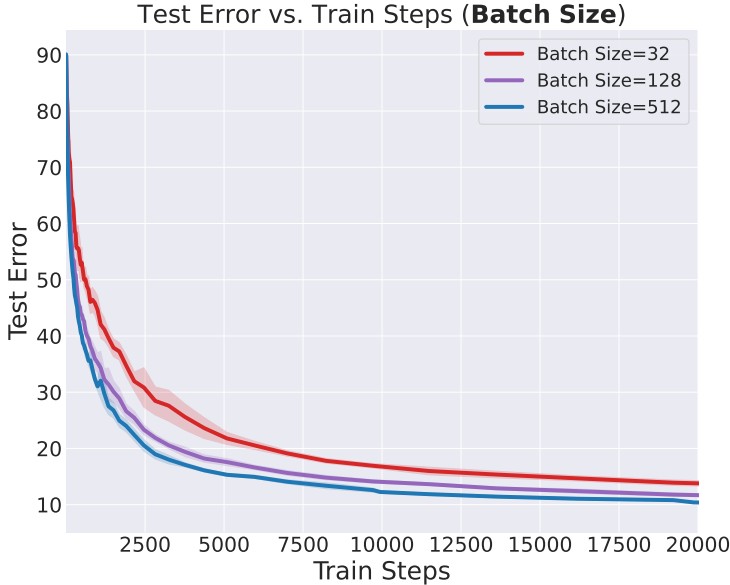

Figure 9: Test performance with error bars for CIFAR-5m in the online setting varying batch size. Note that all figures in Section 2 were already averaged over $\geq 4$ runs.

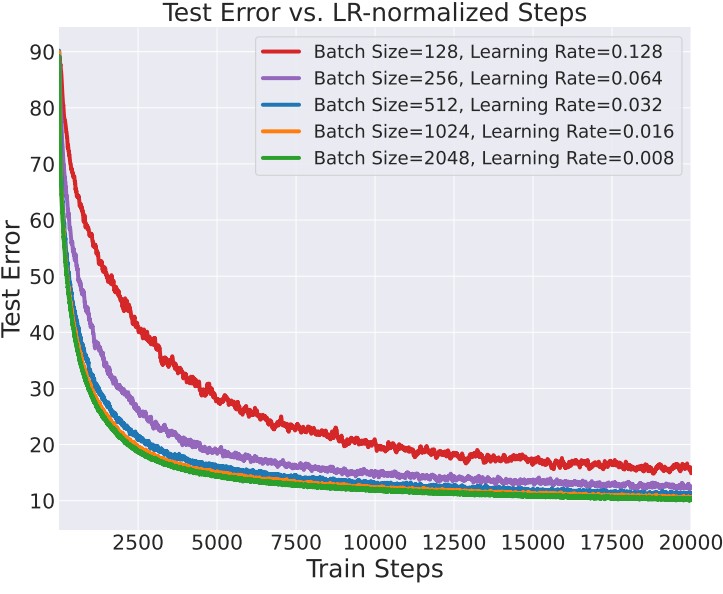

Figure 10: Convergence to gradient flow: we show that as learning rate decreases and batch size increases the loss curves converge (with x-axis as LR-normalized steps). Consistent with our findings, the lower SGD noise runs continue to perform better.

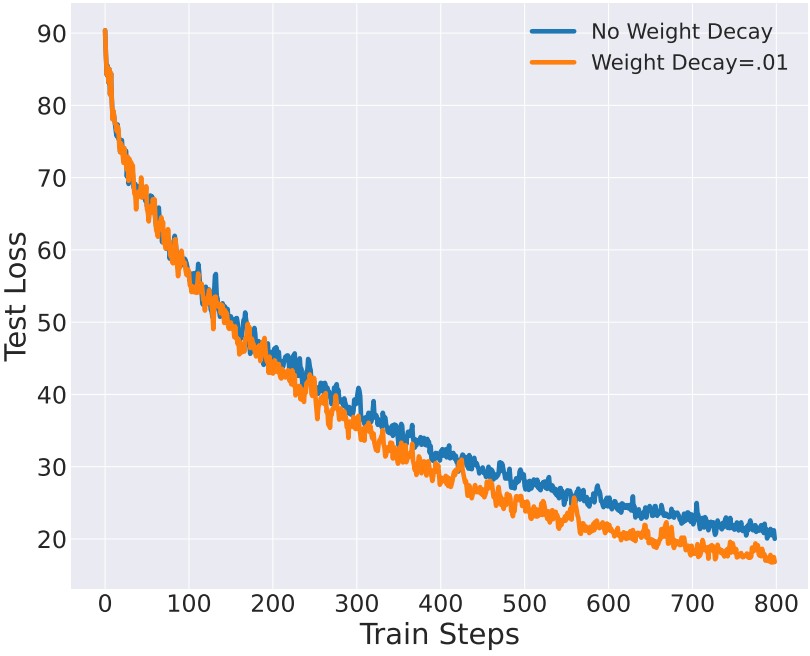

Figure 11: Explicit regularization can help in online learning. Specifically when training on the CIFAR 5m dataset, the test loss for a run using a small amount of weight decay is lower than that of a run with no weight decay.

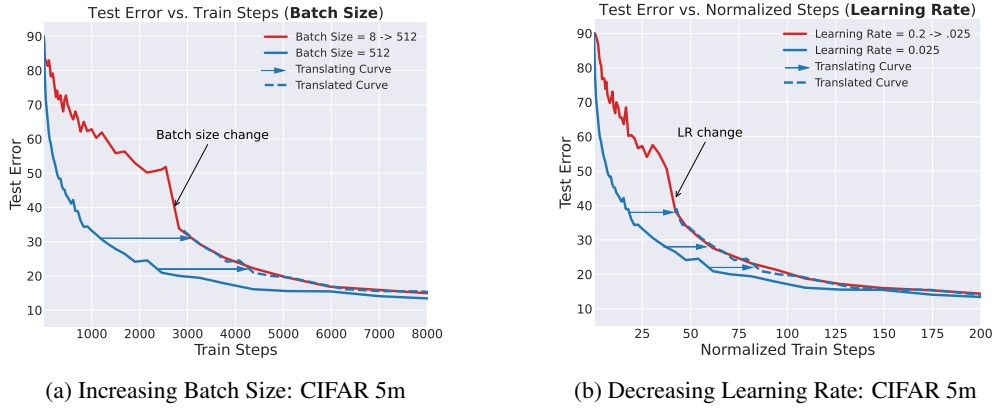

(a) Increasing Batch Size: CIFAR 5m

(b) Decreasing Learning Rate: CIFAR 5m

Figure 12: Decreasing SGD noise (**left**: batch size, **right**: learning rate) during training for CIFAR 5m (**top row**) and C4 (**bottom row**). The red curves correspond to models trained with low SGD noise from initialization, and the blue curves correspond to models trained with high SGD noise and then dropped after $T_0$ steps. Across all experiments, dropping SGD noise causes the blue curves to follow a translated version (dashed red) of the low-SGD noise model curve.

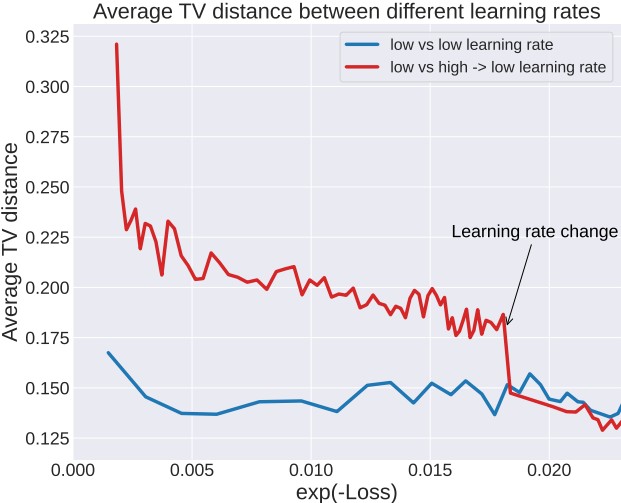

Figure 13: Average TV distance on C4 between models corresponding to the aforementioned runs. Blue corresponds to the baseline with two independent runs with low SGD noise (low learning rate), and red corresponds to the comparison with a low-noise run and a run with a drop from high to low learning rate. Following the LR decay, the red curve drops and reaches a comparable average TV distance than the blue curve. Points on the curves start after models have been trained for 50 (LR-normalized) steps.

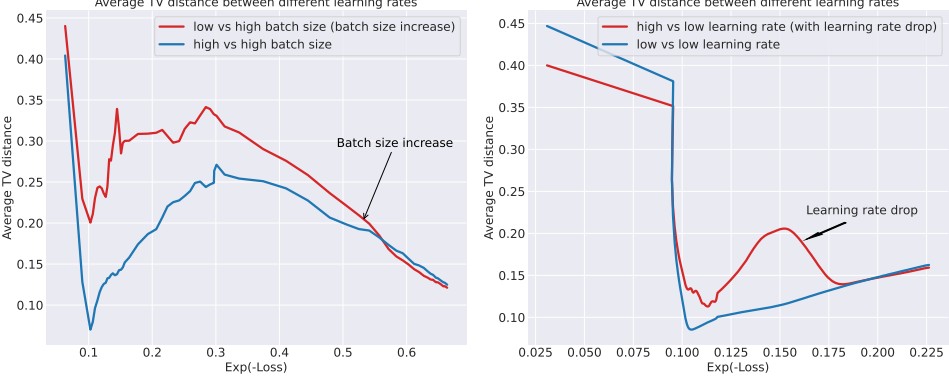

Figure 14: Average TV distance behavior on CIFAR-5m after reducing SGD noise. On the left panel we increase the batch size from 16 to 256 and on the right panel we reduce the learning rate from 0.0512 to 0.0004. The figure demonstrates that the distance between the model with reduced noise and a low-noise model at the same loss is nearly identical to the distance between two random low-noise models trained with the same hyperparameters.

