# OpenReview forum: "Beyond Implicit Bias: The Insignificance of SGD Noise in Online Learning"
_ICLR.cc/2024/Conference — ICLR 2024 Conference Withdrawn Submission_

### Official Review · Reviewer_7b7M · 2023-10-30

**Soundness:** 2 fair
**Presentation:** 2 fair
**Contribution:** 2 fair
**Rating:** 5
**Confidence:** 1

**Summary:**

This paper investigates the influence of SGD noises, specifically batch size and learning rate, on implicit bias within an online learning context. Through comprehensive experiments, the authors demonstrate that unlike in offline settings, SGD noise does not confer any additional advantages in online learning.

Furthermore, the authors introduce and explore the "golden path hypothesis" in relation to online learning. Empirical analysis suggests that for real-world data utilizing deep neural networks, a "noiseless" or "golden" path trajectory may be present, implying that SGD could potentially mimic the trajectory of gradient flow algorithms.

**Strengths:**

The problem studied in this paper is important as the LLMs might adopt the examined online method to update their parameters. This paper performs extensive experiments to support their emperical findings.

**Weaknesses:**

The online learning setting investigated lacks a rigorous and detailed formulation.  See more details in Questions.

**Questions:**

1. The online learning protocol discussed in this paper is not entirely clear to me. Could the authors provide with a more detailed formulation of the online learning procedure using SGD? In the online learning contexts I'm familiar with, such as Prediction with Experts' Advice, regret is typically employed as a performance measure. Could the authors clarify how the algorithm's loss is assessed in the online learning setting under consideration?

2. I'm curious about the relationship between the convergence rate and the choice of adaptive learning rate. Is the observed behavior consistent when using optimizers like Adam?

3. How does this research account for or negate the effects of the neural network's architecture?

4. I'm interested in understanding the design of the experiments. Given that in the real-world online learning setting, achieving comparable performance can be more challenging without full access to the dataset, yet it offers efficiency advantages. Were there particular measures or modifications incorporated to guarantee an fair comparison with the offline setting?

While empirical studies involving SGD algorithms fall outside my primary domain of expertise, I am open to further discussions on the topic.

---

> ### Author Response · Authors · 2023-11-17
> **Response to Reviewer 7b7M**
>
> We thank the reviewer for their comments. We address specific points below:
>
> >The online learning protocol discussed in this paper is not entirely clear to me. Could the authors provide with a more detailed formulation of the online learning procedure using SGD? In the online learning contexts I'm familiar with, such as Prediction with Experts' Advice, regret is typically employed as a performance measure. Could the authors clarify how the algorithm's loss is assessed in the online learning setting under consideration?
>
> The definition of online learning is provided on the first line of Page 2 as single epoch training. In particular, we are considering the *stochastic online learning* setting, where at each instance the loss function is drawn from an underlying distribution, and the regret is measured according to the loss exhibited by the current algorithm and an underlying truth (which is assumed to be providing the correct labels at each step).
>
> > I'm curious about the relationship between the convergence rate and the choice of adaptive learning rate. Is the observed behavior consistent when using optimizers like Adam?
>
> Yes, the experiments in section 2 on Imagenet and C4 dataset are with Adam. The CIFAR-5m experiments use SGD.
>
> >How does this research account for or negate the effects of the neural network's architecture?
>
> This paper shows that the result holds on architectures including CNNs and Transformers, thus showcasing the agnostic nature of the results with respect to architecture.
>
> >I'm interested in understanding the design of the experiments. Given that in the real-world online learning setting, achieving comparable performance can be more challenging without full access to the dataset, yet it offers efficiency advantages. Were there particular measures or modifications incorporated to guarantee an fair comparison with the offline setting?
>
> Note that in this work, we are considering the *stochastic online setting*, and thus the samples are indeed drawn from an underlying distribution even for the online setup, which does not change with time. In this sense, this is not the adversarial online setting.
> Moreover, the only difference between online and offline setting in this work is whether we run training for multiple epochs or not.
>
> Please consider raising your score if you feel that your concerns were answered, and if not we are happy to continue discussing your comments.

---

### Official Review · Reviewer_6xnX · 2023-10-30

**Soundness:** 3 good
**Presentation:** 3 good
**Contribution:** 2 fair
**Rating:** 6
**Confidence:** 3

**Summary:**

This paper studies the potential implicit bias effect of SGD noise in online learning. The authors observe from experiments that in online learning settings. SGD noise does not bring any implicit bias and it is "just noise".  Next, based on experiments, the authors also proposed the "golden path" hypothesis, which states that SGD with different noise levels follows the same trajectory (which they call "golden path") in function space in online learning setting. The authors also perform experiments to support their hypothesis.

**Strengths:**

1. The main result in this paper that SGD follows the same path in online learning settings is an interesting finding in my opinion.

2. The experiments support the main claims well, and the claims made by the paper are stated clearly in general.

**Weaknesses:**

1. I would like to understand more about the scope of the main results:

- The experiments are performed on Resnet18, ConvNext-T, and GPT-2 small, which are relatively large models. I'm wondering if the main hypothesis of this paper also holds for smaller models, or if this phenomenon might be due to the overparameterization of the models?

- The study of this paper focuses on SGD noise, i.e. the noise comes from not using full-batch. I'm wondering if the main hypothesis also holds for manually added noise (e.g. noisy gradient descent like Langevin dynamics) ?

- A minor point: In the paper, your main findings and hypothesis are made for SGD, while in your experiments, the optimizers used are SGD with momentum (for ResNet18), AdamW (for ConvNext-T), and "default optimizer in Mosiac ML" . So the main hypothesis is not only for SGD but also for difference optimizers?

- As you mentioned in the discussion at the bottom of page 2, the "golden path" is the noiseless gradient flow. I'm wondering if you could compare the trajectory of SGD to the actual gradient flow (i.e. GD with full batch, and very small step size) ?

2. The experiments on reducing the step size are not very clear to me, since the step size also affects the sharpness of the solution SGD can find (as you the "edge of stability" phenomenon you mentioned in Appendix D). So it seems to me that the decrease of loss after decreasing the step size may be due to the fact that the dynamic is around a local minimum of certain sharpness, and a smaller step size allows it to go into this local minimum, rather than a better approximation of the "golden path" due to smaller SGD noise. Similar arguments could also made for the experiments on increasing step size.

**Questions:**

Please refer to the strengths and weaknesses part.

---

> ### Author Response · Authors · 2023-11-17
> **Response to Reviewer 6xnX**
>
> We would like to thank the reviewer for their insightful comments. We appreciate the reviewer observing that our paper is interesting and their general curiosity towards the work. We address the comments raised by the reviewer below.
>
> >The experiments are performed on Resnet18, ConvNext-T, and GPT-2 small, which are relatively large models. I'm wondering if the main hypothesis of this paper also holds for smaller models, or if this phenomenon might be due to the overparameterization of the models?
>
> We start by noting that for Resnet18 and GPT-2 small the number of parameters in the model is smaller than the dataset. Further we did experiments on narrower variants of Resnet18 (8 channels instead of 64) as well as small CNNs and found no differences in our results. We would be happy to add these to the final version. We would also be happy to share these in the rebuttal period if the reviewer thinks this is critical to their opinion of the paper.
>
>
> > A minor point: In the paper, your main findings and hypothesis are made for SGD, while in your experiments, the optimizers used are SGD with momentum (for ResNet18), AdamW (for ConvNext-T), and "default optimizer in Mosiac ML" . So the main hypothesis is not only for SGD but also for difference optimizers?
>
>
>
> In general, by SGD noise, we refer to the noise induced due to mini-batch sampling and finite learning rates. Our general claim is about trajectory with and without SGD noise. We would clarify this in the paper that our results hold for other optimizers as well. We note that the MosaicML optimizer is also AdamW.
>
> >The experiments on reducing the step size are not very clear to me, since the step size also affects the sharpness of the solution SGD can find (as you the "edge of stability" phenomenon you mentioned in Appendix D). So it seems to me that the decrease of loss after decreasing the step size may be due to the fact that the dynamic is around a local minimum of certain sharpness, and a smaller step size allows it to go into this local minimum, rather than a better approximation of the "golden path" due to smaller SGD noise. Similar arguments could also made for the experiments on increasing step size.
>
> Yes, we indeed agree that the drop in loss happens both due to Edge of Stability-like phenomena and due to reduction in noise. The main claim is that SGD noise doesn’t affect the overall dynamics in the sense that you can recover the golden path by reducing the noise during training. We are not claiming that the SGD path itself is a noisy version of the gradient flow/golden path.
>
> >The study of this paper focuses on SGD noise, i.e. the noise comes from not using full-batch. I'm wondering if the main hypothesis also holds for manually added noise (e.g. noisy gradient descent like Langevin dynamics) ?
>
>
>
>
> This is an interesting question and our current intuition would be that it would not help in online learning tasks. But we do not have any experiments to this end since our work was focused on SGD noise.
>
>
>
> >As you mentioned in the discussion at the bottom of page 2, the "golden path" is the noiseless gradient flow. I'm wondering if you could compare the trajectory of SGD to the actual gradient flow (i.e. GD with full batch, and very small step size) ?
>
> We do this in:
> 1. Loss trajectory: Figure 10 where we show that with smaller learning rates and larger batch sizes the loss trajectory converges to gradient flow and the performance improves with smaller learning rates and larger batch sizes.
> 2. Function space trajectory: In Figure 14 we decrease the learning rate by more than a factor of 100 and increase the batch size by a factor of 16 and find that after we decrease the SGD noise the TV distance curves match.
>
> We note that the main claim is that SGD noise doesn’t affect the overall dynamics in the sense that you can recover the golden path by reducing the noise during training. We are not claiming that the SGD path itself is a noisy version of the gradient flow/golden path.
>
> Please consider raising your score if you feel that your concerns were answered, and if not we are happy to continue discussing your comments.

---

### Official Review · Reviewer_RXA2 · 2023-11-01

**Soundness:** 2 fair
**Presentation:** 3 good
**Contribution:** 2 fair
**Rating:** 5
**Confidence:** 2

**Summary:**

This paper carries out a series of experiments to compare offline learning (multiple-epoch training) with online (single epoch) learning. The experiments are based on commonly used image and language data and the focus of the experiments is the role played by "SGD noise". High SGD noise refers to high learning rate or small batch size. Unlike the offline learning, the benefits of SGD noise are not observed in the experiments for online learning. It is conjectured that SGD in the online learning case can be interpreted as noisy learning along the "golden path" of the noiseless gradient flow algorithm.

**Strengths:**

1. It is very interesting that SGD noise plays a different role between single and multiple epoch regimes.

2. Figures are well-presented and convey succinct summary of experimental results.

3. The expressions "Fork in the Road" and "Golden Path" are eye-catching terms that create instant curiosity.

**Weaknesses:**

1. The paper is mostly well written; however, the details behind the experimental results are somewhat sparse, including the appendix. Some further clarifications would strengthen the paper substantially. For example, on page 5, it is stated that "To imitate the online regime with ImageNet, we only train for 10 epochs with data augmentation." In the abstract, online learning refers to the single epoch regime but on page 5, it seems that this is not the case. Furthermore, Appendix A contains very short explanations for each of experiments. It is hard to understand exactly what was done in the experiments given the sparse information provided in the paper.

2. All the claims in the paper are entirely driven by the experiments; there are no theoretical results. It would be more prudent if the author(s) could provide the limitation of the current paper on page 9.

**Questions:**

1. It is unclear how many epochs are considered in multiple-epoch training across different experiments. For example, in Figure 1, the top and bottom rows, respectively, show the results from offline learning (multiple-epoch training) and those from online learning (single-epoch training). The training steps are on the same scale between the top and and bottom rows. In the case of top rows, there is no indication of how many epochs are considered. It would be helpful to provide further details.

2. Related to the previous point, is it OK to interpret the X axis the same way between the top and bottom figures in Figure 1? For example,  would it be possible that the patterns observed in offline learning can appear if the number of training steps in online learning is much larger, say, 10 or 100 time larger than 4000? The early paths observed in offline learning are quite similar to those observed in online learning.

3. In addition, what are details of multiple-epoch training? Is multiple-epoch training conducted via random shuffling of the datapoints after each epoch or a simple random sampling of data points with replacement at each step (or something different)? Again it would be helpful to understand the exact nature of multiple-epoch training.

4. The provided supplementary material does not include replication files. Given that the current paper is experimental, it would be useful if all replication files are provided on public domain.

---

> ### Author Response · Authors · 2023-11-17
> **Response to Reviewer RXA2**
>
> We would like to thank the reviewer for their time reviewing our work. We appreciate the reviewer observing that our paper is interesting, well presented and their general curiosity towards the work. We address the comments raised by the reviewer below.
>
> **Details about the experimental results:** We provided these in Appendix A across all three datasets, including details about all pertinent hyperparameters. Please let us know if there is specific information absent in this section; we are happy to discuss them and provide any other details.
>
>
> **Multiple epochs for Imagenet:** Since the Imagenet dataset has fewer samples than the other two datasets we use (CIFAR-5m and C4) to achieve reasonable accuracy we had to train for more than one epoch. To that end, we used data augmentation as a way to remain in the “online regime”. We verified that we were in the online regime by checking that train and test loss/error were the same. We agree this is a valid limitation and will further discuss it in the final version.
>
>
> **Theoretical results:** Yes, we do agree that we don’t have any theoretical results as of now. However, for convex setting, we can show that our arguments hold for specific cases, and we provide an example below. For non-convex settings, it is not difficult to come up with a counter-example in which small learning rate trajectories get stuck in local minima and hence  perform worse. However, as our results hold in practical settings, we believe it would be interesting to come up with the correct assumptions in non-convex settings for which our results hold. Below we describe a simple convex setting where our results hold:
>
> Certainly, here is the text with LaTeX formatting for the mathematical expressions, where each expression is enclosed in $ signs to render as math when compiled with LaTeX:
>
> Consider a 1D convex optimization problem where $ x \sim N(0, 1) $ and $ y^* = 0 $. The square loss is given by $ 0.5w^2x^2 $. For SGD in online learning, with batch size 1 and learning rate $ \eta $, the iterate at time $ t $ is given by $ w_t = \prod_{i=1}^{t} (1 - \eta x_i^2)w_0 $. Then, the expected loss at time $ t $ is given by $ 0.5w_0^2(1 - 2\eta + 3\eta^2)^t $.
>
> Consider another SGD trajectory, with learning rate $ \eta/2 $; its loss at time $ 2t $ is given by $ 0.5w_0^2(1 - \eta + 0.75\eta^2)^{2t} $. Now, if at time $ t $, we switch the first trajectory from $ \eta $ to $ \eta/2 $, then the expected loss at time $ t' > t $ is given by $ 0.5w_0^2(1 - 2\eta + 3\eta^2)^t(1 - \eta + 0.75\eta^2)^{t'-t} $. Thus, in this convex setting, the shape of the expected loss curve will be the same for the two curves, but only after an appropriate time shift to allow for the losses to match.
>
>
> **Unclear how many epochs are in each plot:**  We will add the number of epochs in the caption. For now we give the number of epochs in Fig 3. 3a (top): 15, 60, 240 epochs for batch size 32, 128 and 512 batch size respectively. 3a (bottom): 240, 60, 30 for LRs .025, .1 and .2 respectively. 3b (top): 56, 112, 224 for batch sizes 2048, 4096 and 8192 respectively. 3b (bottom): 168, 84, 56 epochs for LRs .0003, .0006 and .001 respectively.
>
> **Offline curves track online curves:** The reason that offline curves track the online ones is not a coincidence. Rather, every offline training run starts with new data never seen before. Then, as data is repeated more and more, the train and test performance curves diverge and the difference between the regimes becomes noticeable. See Nakkiran et al. 2020 (https://arxiv.org/abs/2010.08127) for a detailed discussion of this phenomenon.
>
> **Effect of longer training in online regime:** In all of our experiments, to the best of our budget, we never saw that the lower noise (with the right movement axis) was performing better than the high noise ones. This includes runs of larger scale than Figure 1.
>
> **Details of multiple-epoch training:** The multi epoch training was done with full shuffle before each epoch.
>
> **Code:** The codebase used for the C4 experiments is LLM foundry: https://github.com/mosaicml/llm-foundry/tree/main. We will add the codebases for CIFAR-5m and Imagenet in a few days.
>
>
> Please consider raising your score if you feel that your concerns were answered, and if not we are happy to continue discussing your comments.